# Possible Roles of Periostin in the Formation of Hemodialysis Vascular Access Stenosis after Polytetrafluoroethylene Graft Implantation in Dogs

**DOI:** 10.3390/ijms21093251

**Published:** 2020-05-04

**Authors:** Kenji Watase, Denan Jin, Kentaro Terai, Taketoshi Kanemiya, Hyogo Nakakura, Nobuhisa Shibahara, Shuji Arima, Shinji Takai

**Affiliations:** 1Department of Innovative Medicine, Graduate School of Medicine, Osaka Medical College, 2-7 Daigaku-machi, Takatsuki, Osaka 569-8686, Japan; kenzzy36@yahoo.co.jp (K.W.); in1381@osaka-med.ac.jp (K.T.); pha010@osaka-med.ac.jp (S.T.); 2Department of Nephrology, Kindai University Nara Hospital, 1248-1 Otodacho, Ikoma, Nara 630-0293, Japan; 3Arisawa General Hospital, 12-14 Higashino-cho, Nakamiya, Hirakata, Osaka 573-1195, Japan; taketoshi78@yahoo.co.jp (T.K.); hyogotwmu@yahoo.co.jp (H.N.); shibahara@arisawahosp.jp (N.S.); 4Division of Nephrology, Department of Internal Medicine, Kindai University Faculty of Medicine, 377-2 Ohno-higashi, Osaka-sayama, Osaka 589-8511, Japan; shuarima@med.kindai.ac.jp

**Keywords:** periostin, PTFE graft, intima, stenosis, hemodialysis

## Abstract

Periostin, a recently found matricellular protein, has been implicated in neointima formation after balloon injury. However, the relationship between periostin and hyperplastic intima formation after PTFE graft implantation is unclear. Under mixed anesthesia, PTFE grafts were implanted between the canine carotid artery and jugular vein, and PTFE graft samples were harvested 1, 2, and 4 months after implantation. Intima formation started on the luminal surface of PTFE grafts at the venous anastomotic region 1 month after implantation. Thereafter, the increase in intimal volume was not only observed in the venous and arterial anastomotic regions, but also in the middle region of the PTFE grafts. In accordance with the increased intimal formation, time-dependent increases in mRNA expressions of periostin and transforming growth factor beta 1 (TGF-β1), as well as a strong positive correlation between periostin and TGF-β1, were observed. These findings suggest that periostin may play a very important role in the pathogenesis of hemodialysis vascular access stenosis through the acceleration of intimal formation. Thus, periostin may be a very important therapeutic target for the treatment of vascular access graft dysfunction in hemodialysis patients.

## 1. Introduction

If kidney function drops to less than 85 to 90 percent of the normal condition in patients with end-stage renal disease (ESRD), these patients will need to have hemodialysis treatment for their whole life unless they are able to obtain a kidney transplant. Advances in hemodialysis techniques have been remarkable, and the survival time on hemodialysis can even reach several decades [1]. Of course, to support efficient hemodialysis in these patients for such a long term, a well-functioning vascular access route is indispensable.

In the clinical setting, a native arteriovenous (AV) fistula in hemodialysis patients as a vascular access route is the first-choice vascular access route due to a lower incidence of complications and better long-term patency [2]. However, the creation of the AV fistula is sometimes restricted due to the lack of suitable vascular sites, especially in diabetic patients with ESRD [2]. If a fistula is not an option for patients, the next best type of hemodialysis access is an AV graft, namely, a surgically created artificial conduit that connects an artery to a vein as a vascular access route. Currently, artificial polytetrafluoroethylene (PTFE) conduits are the most commonly used vascular prostheses in the clinical setting. However, primary patency of the AV PTFE graft is not ideal. It has been reported that about 40–50% of hemodialysis patients with PTFE grafts needed percutaneous transluminal angioplasty (PTA) or reconstruction of another vascular access route within 1 year due to vascular access dysfunction [3]. Therefore, the health insurance costs needed to cover such treatments have become a major problem. Consequently, improvements in vascular access methods are urgently needed.

With respect to PTFE vascular access graft dysfunction, it has been recognized that the stenosis caused by intimal hyperplasia on the venous side is the main hemodynamic cause [3]. Although the precise mechanisms that lead to intimal formation in the AV PTFE graft remain unclear, it has been suggested that several factors, such as the occurrence of vessel stretch and shear stress, as well as the involvement of some well-known growth factors, i.e., platelet-derived growth factor (PDGF) [4], transforming growth factor beta 1 (TGF-β1) [5], and angiotensin II [6], participate in this dysfunction. On the other hand, despite great efforts by many scientists to elucidate the pathogenetic mechanisms of the vascular access graft dysfunction caused by intima formation in the AV PTFE graft, effective therapeutic methods for vascular access graft dysfunction have not yet been found. Therefore, another pathophysiological mechanism may be involved.

Recently, periostin, a newly identified extracellular matrix (ECM) protein, was reported to be important in the promotion of several types of vascular remodeling, such as that involving the carotid artery intima in rats [7], atherosclerotic lesions in mice [8], and coronary artery-injured intima in swine [9]. Periostin is a 90-kDa secreted ECM/matricellular protein, and it has been reported that periostin and TGF-β1 regulate the expression of each other under pathophysiological conditions [10,11]. It is also well known that TGF-β1 is a multi-function polypeptide that is very important in the regulation of phenotypic transformation in smooth muscle cells (SMCs) [12] and mesenchymal cells [13], and this phenotypic transformation is known to be the pathological foundation of intima formation. Interestingly, we have found that expression of TGF-β1 in the tissue surrounding the artificial PTFE vessel and the proliferative intima was significantly increased during both acute and late phases after the implantation of an artificial PTFE vessel in dogs [14]. All of the above evidences suggest that periostin, through its interaction with TGF-β1, may participate closely in the pathogenesis of vascular access graft dysfunction caused by intima formation in AV PTFE grafts. In the present study, the detailed time-dependent changes of periostin expression, as well as the relationship between periostin and TGF-β1, were examined using a canine artificial PTFE vessel implantation model.

## 2. Results

### 2.1. The Time-Dependent Intima Formation at 1, 2 and 4 Months after PTFE Graft Implantation

As can be seen in Figure 1A, apart from the inner luminal surface of the PTFE graft at the venous anastomotic region, intima formation was not very evident 1 month after PTFE graft implantation. Two months later (Figure 1B), intimal formation was more notable on the inner PTFE graft luminal surface at the venous anastomotic region, and this intima also appeared on the middle region of the PTFE graft luminal surface. Four months later (Figure 1C), intima formation on the inner luminal surface of PTFE grafts at all three regions became more evident. Figure 1D shows the intima areas calculated by a computerized morphometry system. As can be seen in these bar graphs, total intima formation was time-dependently increased until 4 months after PTFE graft implantation. Furthermore, analysis of the three different individual regions of the PTFE grafts (i.e., venous, middle, and arterial regions) showed that the intima formation at the arterial and middle regions happened rarely in the early phase (1 month), and their intimal volumes were increased during the later phase (2–4 months) after PTFE graft implantation. On the other hand, intimal formation was faster at the venous region of the grafts than at other regions, reaching a peak at 2 months and then maintaining these volumes thereafter.

### 2.2. The Time-Dependent Expression of Periostin 1, 2, and 4 Months after PTFE Graft Implantation

At 1 month after PTFE graft implantation (Figure 2A), periostin immunostaining was found at the intimal region of both arterial and venous regions, as well as at the tissues surrounding the PTFE graft. In parallel with the time-dependent increase in the intima area, the areas of periostin-positive staining were significantly increased either in the inside of the intima or the outside of the tissues surrounding the PTFE graft (Figure 2B,C). Like the periostin immunostaining, periostin mRNA expression was detectable as early as 1 month after PTFE graft implantation, and these expressions were more marked 4 months later (Figure 2D) at both regions examined.

### 2.3. The Time-Dependent Expression of TGF-β1, 1, 2, and 4 Months after PTFE Graft Implantation

Like the periostin immunostaining, TGF-β1-positive staining was also found in the intima of both arterial and venous regions, as well as at the tissues surrounding the PTFE graft 1 month after PTFE graft implantation (Figure 3A). A similar immunostaining pattern for TGF-β1 was also found 2 and 4 months after PTFE graft implantation (Figure 3B,C). As can be seen in Figure 1D, TGF-β1 mRNA expression was detectable as early as 1 month after PTFE graft implantation, and these expressions were significantly increased 4 months later compared to the 1-month post PTFE graft implantation time point (Figure 3D).

### 2.4. The Components of the Intima after PTFE Grafts Implantation

Figure 4A shows the immunohistologic examinations of the serial sections from the venous anastomotic regions with antibodies against vimentin and *α*-SMA. These loci were matched to the yellow frames of Azan–Mallory staining, as shown in Figure 4B. The intima areas in the venous anastomotic region were occupied mainly by *α*-SMA-positive cells, and the immunostaining for vimentin in the serial section showed that most of these *α*-SMA-positive cells were overlapped with the vimentin-positive cells. On the other hand, as can be seen in the high magnification views of vimentin and *α*-SMA immunostaining, although vimentin-positive staining was found in all cellular components at these venous regions (including either in the intima area and in the splits of PTFE material wall), some cellular components in the border zone between the intima and the PTFE material wall, as well as in the splits of the PTFE material wall, were negative for *α*-SMA (yellow arrows, Figure 4A). A blue color on Azan–Mallory staining indicates collagen distribution in the venous anastomotic region, suggesting that collagen also constitutes part of the intima component. Figure 4C shows the expression of collagen I mRNA in three different regions 1, 2, and 4 months after PTFE graft implantation. Collagen I expressions were time-dependently increased in all three regions.

### 2.5. Linear Regression Analyses

In the present study, linear regression analysis was also performed to investigate the relationships among expressions of periostin, TGF-β1, and collagen I. As indicated in Figure 5, a strong positive correlation between the expression of periostin and TGF-β1 was observed, independent of time point and region. A similar correlation between the expressions of TGF-β1 and collagen I, as well as between the expressions of periostin and collagen I, was also found (Figure 5).

Figure 6 shows the relationships among the expressions of periostin, TGF-β1, and collagen I at different time points post-PTFE graft implantation. The correlations among the expression levels of these factors were analyzed at 1, 2 and 4 months after PTFE graft implantation, irrespective of the region to which the data belonged. As can be seen in these figures, the correlations among the three factors were strong at 1 and 2 months after PTFE graft implantation, and these correlations became weaker thereafter. As can be seen in Figure 6, at the post 4-month PTFE graft implantation time point, the calculated *p* values became greater and one of the *p* values was no longer significant.

Figure 7 shows the relationships among the expressions of periostin, TGF-β1, and collagen I at the three different regions. The correlations among the expression levels of these factors were analyzed at the arterial, middle, and venous regions after PTFE graft implantation, independent of the time point. Significant correlations between the expressions of TGF-β1 and collagen I, as well as between the expressions of periostin and TGF-β1, were observed at all three examined regions. However, a correlation between the expressions of periostin and collagen I was observed only at the venous anastomotic region.

## 3. Discussion

In the present study, in accordance with the time-dependent increase in the intima volume, periostin expression in the hyperplastic intima, as well as in the tissue surrounding the PTFE graft, was significantly increased after implantation of artificial PTFE vessels in dogs. A close correlation between the expressions of periostin and TGF-β1 was also seen. On the other hand, the intima areas in the venous anastomotic region were occupied mainly by *α*-SMA-positive cells, and the immunostaining for vimentin in the serial section showed that most of these *α*-SMA-positive cells were overlapped with the vimentin-positive cells, indicating that these cells originated from mesenchymal progenitor cells [15]. As is well known, mesenchymal cells contain mainly fibroblasts with the phenotypic type of myofibroblasts, and since the majority of intima component cells were *α*-SMA-positive, it was concluded that the largest cellular components in the proliferative intima after implantation of PTFE grafts in dogs were myofibroblasts [16]. On the other hand, as can be seen in the high magnification views of vimentin and *α*-SMA immunostaining, although vimentin-positive staining was found in all cellular components at these venous regions (including either in the intima area and in the splits of PTFE material wall), some cellular components in the border zone between the intima and the PTFE material wall, as well as in the splits of PTFE material wall, were negative for *α*-SMA (yellow arrows, Figure 4A), indicating that these cells were fibroblasts. Therefore, myofibroblasts were a major cellular component, and fibroblasts comprised only a small proportion in the hyperplastic intima. Although fibroblasts seemed not likely to be a major cellular component within the hyperplastic intima, our previous studies showed that these cells were important to migrate from one place to others [17,18].

All of these data, taken together with recent accumulated evidence concerning periostin [7,8,9] and TGF-β1 [5,12], suggest that periostin through its interaction with TGF-β1 expression may participate closely in the pathogenesis of PTFE graft stenosis, which is known as a main cause of the development of vascular access graft dysfunction in hemodialysis patients receiving PTFE graft implantation.

Up to now, a relationship between periostin and vascular proliferative disease has been shown previously in the rat and pig balloon-injured models [7,9]. For example, the mRNA expression of periostin in the proliferative neointima was increased significantly after the arterial vessels were mechanically injured by a balloon catheter [7]. Interestingly, overexpression of periostin in SMCs or murine embryonic fibroblast cells could facilitate migration of these cells, which is an essential cellular process leading to vascular intimal formation [19]. That research also found that the migrative response of periostin was mediated through stimulation of the integrins α*ν*β3 and α*ν*β5, with subsequent intracellular signal activation, such as focal adhesion kinase (FAK) phosphorylation, indicating that these integrins act as periostin receptors to modulate cell migration. In the present study, periostin expression determined by RT-PCR showed that mRNA periostin expression could be observed as early as 1 month after PTFE graft implantation at the three examined regions and was further increased 4 months after PTFE graft implantation (Figure 2D). Immunohistochemical examinations showed that periostin was expressed widely not only in the proliferative intima, but also in the tissues surrounding the PTFE graft (Figure 2). The main cellular component was found to be mesenchymal origin cells such as fibroblasts, with the specific phenotype of myofibroblasts (Figure 4), suggesting that the periostin deposited in the intimal proliferative areas originated from secretion by local fibroblasts and myofibroblasts. Indeed, it has been reported previously that periostin is produced by dermal fibroblasts under cytokine stimulation [11]. Interestingly, our previous studies showed that intima formation on the inside surface of the middle PTFE graft seemed to be initiated by migration and proliferation of fibroblasts of external origin [14,17]. Taken together with the above evidence, this indicates that periostin plays a crucial role in the modulation of intimal formation after PTFE graft implantation.

In the present study, the pattern of TGF-β1 mRNA expression was extremely similar to that of periostin (Figure 3D). As shown in Figure 3, TGF-β1-positive staining was also found at the intima region of both arterial and venous regions, as well as at the tissues surrounding the PTFE graft from the early to late phases. Involvement of TGF-β1 in the formation of neointima was reported by Shi et al. using a porcine model of balloon-overstretch coronary artery injury [20]. They found that fibroblast-like TGF-β1-positive cells appeared at the adventitial layer 2 days after injury, and these cells were phenotypically changed to myofibroblast-like TGF-β1-positive cells. At 14 days after injury, the decrease in TGF-β1 coincided with the disappearance of adventitial myofibroblasts, whereas the neointima showed longer TGF-β1 expression. Apart from an important role of TGF-β1 in the promotion of fibroblast transdifferentiation, fibroblast proliferation also seemed to be very important for TGF-β1 stimulation [21]. These transdifferentiative and proliferative roles of TGF-β1 were reported to be mediated through the Smad 2/3 to Akt cascade and autocrine production of FGF-2, respectively [13,22].

In the present study, a strong positive correlation between the expressions of periostin and TGF-β1 was observed independent of time point and region (Figure 5), indicating an intimate relationship between the molecules. Indeed, several papers demonstrated that TGF-β1 induces periostin gene expression, and increased periostin, in turn, promotes TGF-β1 gene expression [22,23]. Moreover, cross-talk between TGF-β1 and periostin has recently been confirmed [24], indicating that increased periostin and TGF-β1 through the promotion of individual molecular production further amplifies their gene expressions, and increases in the individual molecules through their cross-talk may result in very severe pathologic responses.

In the present study, it was not possible to investigate if inhibition of periostin in this model could suppress intimal formation after implantation of PTFE grafts due to the lack of a low-molecular weight periostin inhibitor. Since these investigations are important to confirm that periostin has a direct role in intimal pathogenesis after implantation of PTFE grafts, a final conclusion concerning periostin’s direct involvement in intimal formation in PTFE grafts will have to wait until a periostin-specific inhibitor is tested in this model.

## 4. Materials and Methods 

### 4.1. Surgery and Sample Preparation

Eighteen male beagle dogs weighing 9–12 kg were obtained from Japan SLC (Shizuoka, Japan). Anesthesia was induced in the dogs by intramuscular (IM) injection of medetomidine (0.02 mg/kg) and midazolam (0.3 mg/kg), followed by an intravenous (IV) injection of pentobarbital sodium (10 mg/kg) 10 min later. To avoid blood clotting during and after the surgery, heparin (100 U/kg) was subcutaneously injected after the anesthesia procedure. Under sterile conditions, the PTFE grafts with an internal diameter of 0.6 cm and length of 7 to 8 cm (Atrium Medical Corporation, Hudson, NJ, USA) were implanted between the carotid artery and jugular vein with arterial-graft and venous-graft side-to-end anastomoses. Finally, the skin was closed in layers, and the animals were allowed to recover. All animals received antibiotic therapy through an IM injection of cefminox sodium (0.5 g). Graft patency was monitored weekly by auscultation and palpation. The experimental procedures for animals were conducted in accordance with the guidelines of Osaka Medical College for medical experiments, and they were approved by the ethics committee (29003).

The animals were sacrificed at 1 (*n* = 6), 2 (*n* = 6), and 4 (*n* = 6) months post-implantation; their grafts, together with the connected native arteries and veins, were then carefully removed. All the samples were cut into three equal parts (average length: 2.3–2.6 cm): arterial anastomotic, middle, and venous anastomotic regions, as described previously [17], and the vessels were fixed in methanol-Carnoy’s fixative overnight and then embedded in paraffin.

### 4.2. Histological Examination

In order to obtain wide information about the intima formed not only in the entire inner luminal surface of the PTFE graft but also in the native artery and vein near the respective anastomotic regions after PTFE graft implantation, samples were cut lengthwise to include both the native artery and vein, as reported previously [17]. Once these sections (3 μm in thickness) were obtained, the following general histological and immunohistological examinations were performed.

Azan–Mallory staining was used to determine intimal areas and collagen distribution. 

Immunohistological examinations were performed using a goat anti-human periostin antibody (sc-49480, Santa Cruz, CA, USA; 1:200 dilution), a rabbit anti-human TGF-β1 antibody (21898-1-AP, PROTEINTECH JAPAN, Tokyo, Japan; 1:300 dilution), a mouse anti-human vimentin antibody (M0725, Dako, Carpinteria, CA, USA; 1:70 dilution), and a mouse anti alpha-smooth muscle actin antibody (α-SMA, M0851, Dako; 1:200 dilution), according to the protocol described elsewhere [14,17,18]. In brief, to suppress endogenous peroxidase activity and nonspecific binding, the deparaffinized sections were incubated with 3% hydrogen peroxide and Serum Free Protein Block (X0909; Dako, Glostrup, Denmark) for 5 min at room temperature, respectively. Then, these sections were incubated with the above diluted primary antibodies overnight at 4 °C, followed by reaction with components from a labeled streptavidin-biotin peroxidase kit (Dako LSAB kit, Dako, Carpinteria, CA, USA) that included 3-amino-9-ethylcarbazole color development. Sections were then lightly counterstained with hematoxylin. These histological images were photographed using a computerized morphometry system (NIS-Elements Documentation, version (v.) 3.07, Nikon, Tokyo, Japan).

### 4.3. Real-Time Polymerase Chain Reaction (RT-PCR)

To extract the RNA of the PTFE grafts, 10 sheets of the 20-μm-thick paraffin sections were collected from the respective Carnoy-fixed, paraffin-embedded tissue blocks using a microtome (LITORATOMU, REM-710, Yamto Koki Kogyo Ltd., Saitama, Japan). Total RNA was extracted in accordance with the protocol provided in the total RNA isolation kit (ISOGEN PB Kit, NIPPON GENE CO., LTD., Tokyo, Japan).

Total RNA (1 μg) was transcribed into cDNA with SuperScript VILO (Invitrogen, Carlsbad, CA, USA). Then, mRNA levels were measured by RT-PCR on a Stratagene Mx3000P (Agilent Technologies, San Francisco, CA, USA) using Taq-Man fluorogenic probes. RT-PCR primers and probes for periostin, TGF-β1, collagen I, and GAPDH were designed by Roche Diagnostics (Tokyo, Japan). The primers were as follows: 5′-ttcctgattctgccaaacaa-3′ (forward) and 5′- gtccgtgaaagtggtttgct-3′ (reverse) for periostin, 5′-cattaacgggttcagttccag-3′ (forward) and 5′-agcaggaagggtcggttc-3′ (reverse) for TGF-β1, 5′-gtctgcccggtgaaagag-3′ (forward) and 5′-cagtagcaccatcgtttcca-3′ (reverse) for collagen I, and 5′-aatgtatcagttgtggatctgacc-3′ (forward) and 5′-gcttcactaccttcttgatgtcg-3′ (reverse) for GAPDH. The probes were as follows: 5′- ctggctgg-3′ for periostin, 5′-ggccacca-3′ for TGF-β1, 5′-cctggagc-3′ for collagen I, and 5′-cctggaga-3′ for GAPDH.

The mRNA levels of periostin, TGF-β1, and collagen I were normalized to that of GAPDH.

### 4.4. Histologic Quantitative Analysis and Statistical Analysis

The Azan–Mallory-stained sections obtained from three different regions were used to determine the intimal areas using a computerized morphometry system (NIS-Elements Documentation, version (v.) 3.07, Nikon, Tokyo, Japan). First, the intimal areas covering the grafts were measured. To normalize the observed intimal areas, the lengths of the prosthetic vessels were also measured. The degree of neointima formation in each graft was expressed as the respective area divided by the length of the graft.

All numerical data shown in the text are expressed as means ± standard error of the mean (S.E.M.). Significant differences among the mean values of multiple groups were evaluated by one-way analysis of variance (ANOVA) followed by a post hoc analysis (Fisher’s test). Pearson’s correlation coefficient was measured to test the linear relationship between two variables using linear regression analysis. *p* < 0.05 was considered significant.

## 5. Conclusions

The present study showed that periostin expression in the hyperplastic intima, as well as in the tissues surrounding the PTFE graft, was significantly increased after artificial PTFE graft implantation, and a close correlation between the expressions of periostin and TGF-β1 was found, independent of time point and region. These findings suggest that periostin may play a very important role in the pathogenesis of hemodialysis vascular access stenosis through the acceleration of intima formation; thus, periostin may be a therapeutic target for the treatment of vascular access graft dysfunction in hemodialysis patients.

## Figures and Tables

**Figure 1 ijms-21-03251-f001:**
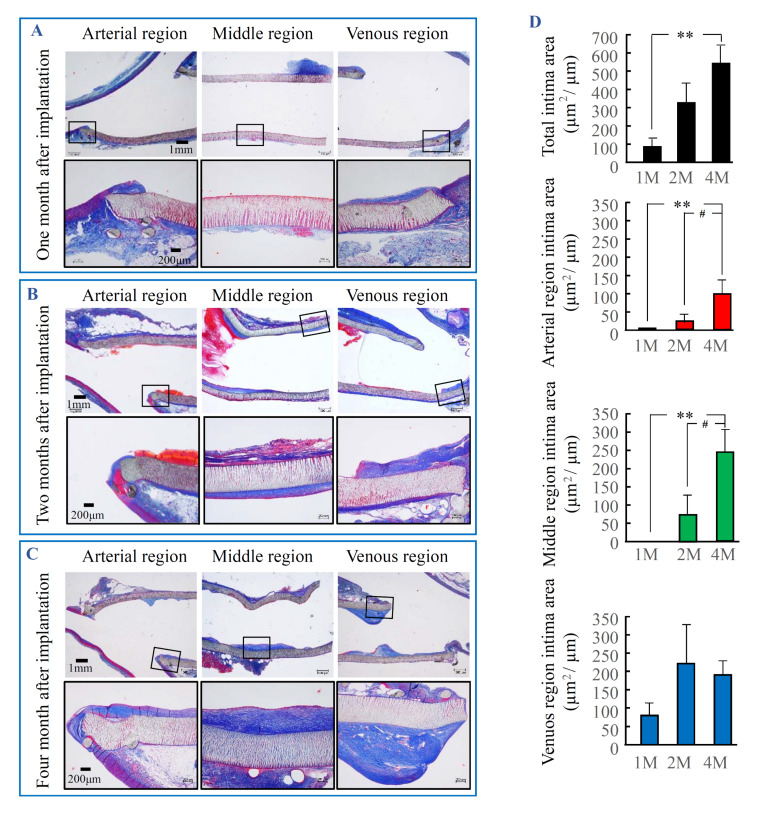
Azan–Mallory staining and the estimated intima areas after PTFE graft implantation. (**A**–**C**): Intimal formation 1, 2, and 4 months after PTFE graft implantation. Graft samples were cut along a lengthwise axis so that sections included both the native artery and vein for histological examination. The black frame of bottom pictures (A, B, C) are the extended image of the black boxes in the upper pictures. (**D**) Time-dependent changes of intima areas calculated by a computerized morphometry system. Each bar represents the mean ± S.E.M. of six grafts. ** *p* < 0.01 vs 1 M; # *p* < 0.05 vs 2 M.

**Figure 2 ijms-21-03251-f002:**
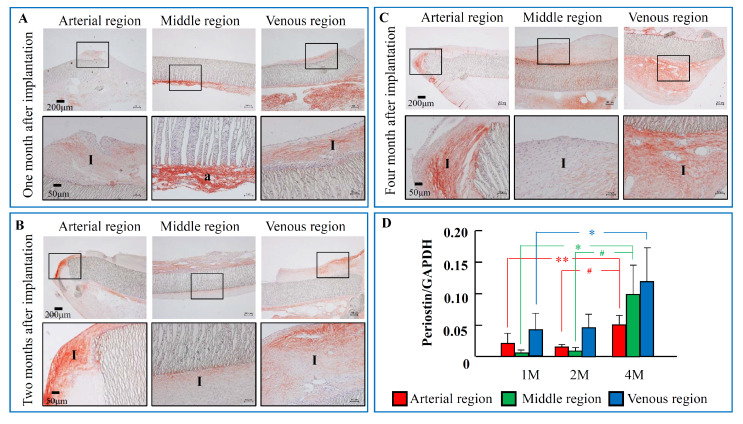
Immunostaining pattern and mRNA expression level of periostin during the 4-month post-PTFE graft implantation period. (**A**–**C**) Representative periostin immunostaining at the different regions of PTFE grafts 1, 2, and 4 months after PTFE graft implantation. The black frame of bottom pictures (**A**–**C**) are the extended image of the black boxes in the upper pictures. (**D**) Time-dependent changes of periostin mRNA expression levels 1, 2, and 4 months after PTFE graft implantation. I: intima side; a: adventitial side. Each bar represents the mean ± S.E.M. of six grafts. * *p* < 0.05, ** *p* < 0.01 vs. 1 M; # *p* < 0.05 vs. 2 M.

**Figure 3 ijms-21-03251-f003:**
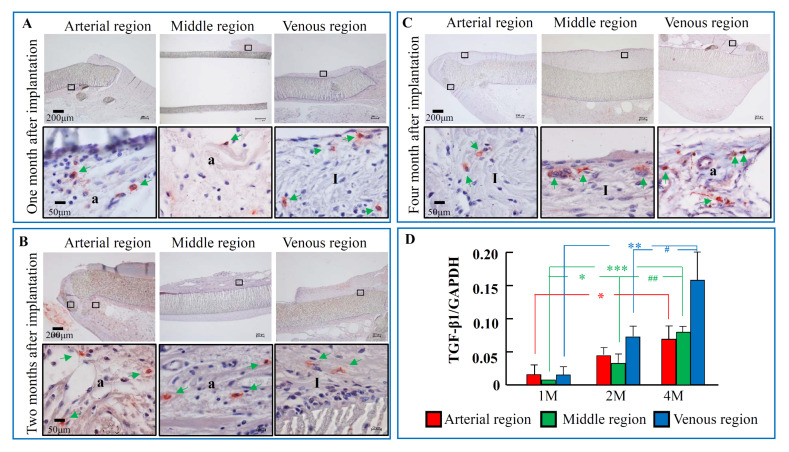
Immunostaining pattern and mRNA expression level of TGF-β1 during the 4-month post-PTFE graft implantation period. (**A**–**C**) Representative TGF-β1 immunostaining at the different regions of PTFE grafts 1, 2, and 4 months after PTFE graft implantation. (**D**) Time-dependent changes of TGF-β1 mRNA expression levels 1, 2, and 4 months after the PTFE graft implantation. I: intima side; a: adventitial side. Green arrows: TGF-β1-positive cells. Each bar represents the mean ± S.E.M. of six grafts. * *p* < 0.05, ** *p* < 0.01, *** *p* < 0.001 vs. 1 M; # *p* < 0.05, ## *p* < 0.01 vs. 2 M.

**Figure 4 ijms-21-03251-f004:**
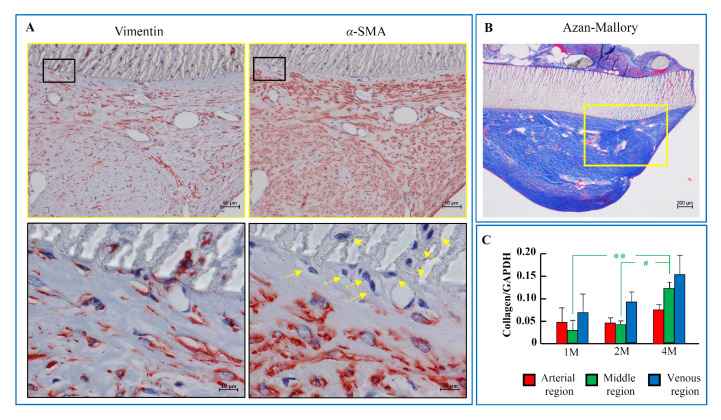
Representative venous immunostaining patterns of vimentin and α-SMA, as well as collagen mRNA expression levels, after PTFE graft implantation. (**A**) Immunohistologic examinations of the serial sections from the venous anastomotic regions with antibodies against vimentin and *α*-SMA. Yellow arrows: *α*- SMA negative cells (**B**) Azan–Mallory staining of the venous anastomotic region. The yellow frames in the Azan–Mallory staining indicate the vimentin and *α*-SMA immunostaining shown in part A. (**C**) Expression of collagen I mRNA in three different regions 1, 2, and 4 months after PTFE graft implantation. Each bar represents the mean ± S.E.M. of six grafts. ** *p* < 0.01 vs. 1 M; # *p* < 0.05 vs. 2 M.

**Figure 5 ijms-21-03251-f005:**
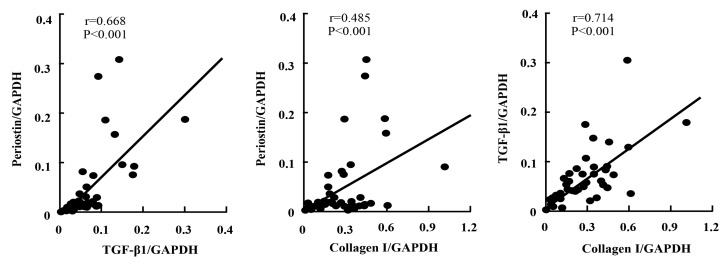
Linear regression analyses of the correlations among the expression levels of periostin, TGF-β1, and collagen I after PTFE graft implantation, independent of time point and region.

**Figure 6 ijms-21-03251-f006:**
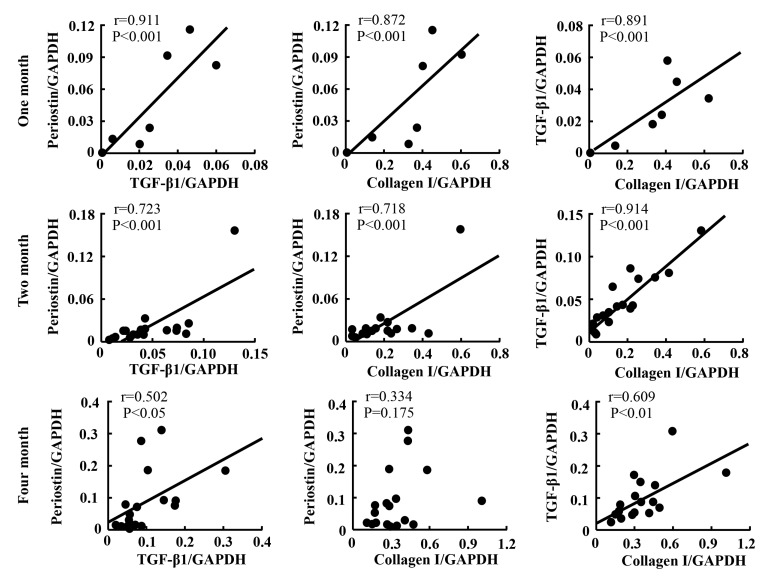
Linear regression analyses of the correlations among the expression levels of periostin, TGF-β1, and collagen I 1, 2, and 4 months after PTFE graft implantation, independent of region.

**Figure 7 ijms-21-03251-f007:**
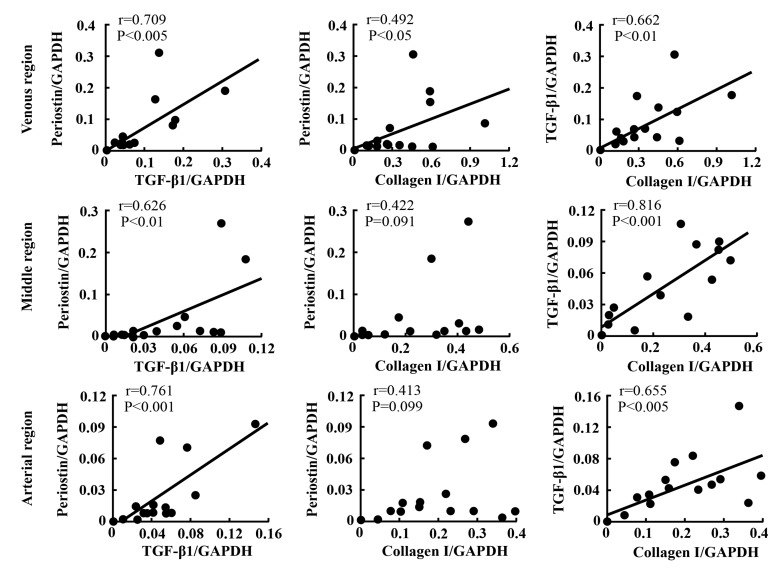
Linear regression analyses of the correlations among the expression levels of periostin, TGF-β1, and collagen I at arterial, middle, and venous regions after PTFE graft implantation, independent of time point.

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
