# Peer review of "Possible Roles of Periostin in the Formation of Hemodialysis Vascular Access Stenosis after Polytetrafluoroethylene Graft Implantation in Dogs"

_ijms, 2020, doi:10.3390/ijms21093251_

Round 1
Reviewer 1 Report
The authors have investigated the role of Periostin in hyperplastic intima formation post PTFE graft implantation in canines. The authors have reported increased intima formation with peak intima formation at 2 months post PTFE graft implantation and intima formation was faster in venous regions of the grafts. The authors have also reported increased expression of Periostin and TGF-β1 post PTFE graft implantation and reported close correlation between TGF-β1 and Periostin independent of time and region of the grafts. It is interesting study but there are few concerns as listed below. Major concerns: 1) Did authors perform mechanistic studies to delineate the mechanism of Periostin in hyperplastic intima formation post PTFE graft implantation and could directly implicate interactions between TGF-β1 and Periostin using inhibitors as driving mechanism for pathogenesis of vascualr access stenosis? 2) Did authors examine the effect of gender in the current study? The females were reported to have poorer outcomes compared to males? Could authors please comment on rational for inclusion of male dogs only in the study? Minor concerns: 1) Please replace "implanteation" with "implantation" on the Y-axis labels of Figure 1A-1C, 2A-2C, 3A-3C. 2) Could authors please show which groups are being compared in the figure to denote statistical significance such as 1 month vs 4 month or 2 months vs four months. It would be easier for readers to interpret the results from the figure rather than referring to the Figure legend for Figure 1D, 2D, 3D, and 4C. 3) Could authors please clarify whether the authors meant to denote the levels of Periostin mRNA expression in arterial region at 4 months vs. one month as *P<0.05 or **P<0.01? Please review and correct the sign used to denote statistical significance in Figure 2D accordingly. 4) Please include details on what would I and A stand for in Figure 2A-C? 5) Please replot to adjust the bar width or decrease the sign sizes to accommodate the signs in correct orientation (Horizontal rather than vertical orientation to avoid confusion to the readers) in Figure 2D, 3D, and 4C. 6) Please replace "collagen I at different post-PTFE graft implantation time points" with "collagen I at different time points post-PTFE graft implantation" on Ln # 156.Author Response
May 01, 2020
International Journal of Molecular Sciences
Editorial Office
Dear Dr. Reviewer 1:
Thank you very much for your E-mail of April 23 regarding our manuscript “Possible roles of periostin in the formation of hemodialysis vascular access stenosis after polytetrafluoroethylene graft implantation in dogs” (Manuscript ID: ijms-787151).
Below please find our point-by-point responses to the reviewers’ comments, and the changes made in the manuscript are highlighted by the “Track Changes” function in Microsoft Word.
Our responses to you is as follows.
Comments and Suggestions for Authors 1
Major concerns: 1)
As the reviewer noted, if inhibition of periostin could suppress intimal formation in this model, this would be very important evidence suggesting that periostin has a direct role in the pathogenesis of vascular access stenosis. However, as we know, it is difficult to obtain a low-molecular weight periostin inhibitor at the present time. Therefore, we were unable to examine the effects of periostin inhibition on intimal formation in this model. Since testing the effects of periostin inhibition on intimal hyperplasia after implantation of PTFE grafts is important to demonstrate a direct role of periostin, we have added this issue as a limitation to the Discussion section as follows (Lines 243-248):
In the present study, it was not possible to investigate if inhibition of periostin in this model could suppress intimal formation after implantation of PTFE grafts due to the lack of a low-molecular weight periostin inhibitor. Since these investigations are important to confirm that periostin has a direct role in intimal pathogenesis after implantation of PTFE grafts, a final conclusion concerning periostin’s direct involvement in intimal formation in PTFE grafts will have to wait until a periostin-specific inhibitor is tested in this model.
Major concerns: 2)
As the reviewer noted, the results of experiments in the present study may differ by sex. It has been recognized commonly that estrogens and androgens have opposite effects in the pathogenesis of hypertrophic vascular diseases (Si et al. Gender difference in cytoprotection induced by estrogen on female and male bovine aortic endothelial cells. Endocrine. 2001 15:255-262.; Kim et al. Carotid intima-media thickness is associated with increased androgens in adolescents and young adults with classical congenital adrenal hyperplasia. Horm Res Paediatr. 2016 85:242-249). Therefore, intimal formation might also be affected by the cyclic changes in sex hormone levels. In the present study, to avoid the variation between females and males, we decided to use only male dogs to exclude the effects of estrogen fluctuation during menstruation in female dogs. Of course, investigation of both sexes may more closely reflect the human pathophysiology after implantation of PTFE grafts.
Minor concerns:
- I apologize for the typographical error and thank the reviewer for pointing it out. We have corrected "implanteation" to "implantation" on the Y-axis labels of Figures 1 to 3.
- As noted, the significance of the results may be better presented directly in the figure itself than in the figure legend. Therefore, we have shown the statistical comparison between groups by lines instead of symbols in the figure.
- We have reconfirmed our original data for periostin mRNA expression in the arterial region at 4 months and at 1 month. All statistically significant results were correct. The value of periostin mRNA expression in the arterial region at 4 months was lower than that of the value of the venous region, and the p values of the 2 groups seem somewhat However, we did not find any mistakes in these results. The greater statistical significance observed in the arterial region at 4 months than in other regions may have been caused by small variations in the arterial region.
- As suggested, we have defined I and A in Figure 2 as follows: I, intimal side; A, adventitial side.
- As suggested, we have modified the bar graphs in Figures 2D, 3D, and 4C.
- As suggested, we have changed “collagen I at different post-PTFE graft implantation time points” to “collagen I at different time points post-PTFE graft implantation”.
We appreciate the helpful comments we received and believe that all these corrections have made our paper much more valuable.
Thank you very much for your kind consideration.
We look forward to the appearance of our revised manuscript in International Journal of Molecular Sciences.
Sincerely yours,
Denan Jin, M.D., Ph.D.
Department of Innovative Medicine, Graduate School of Medicine, Osaka Medical College, 2-7 Daigaku-machi, Takatsuki, Osaka 569-8686, Japan.
TEL: +81-72-683-1221 (Ext2141)
FAX: +81-72-684-6730
E-mail: pha012@osaka-med.ac.jp

Reviewer 2 Report
This manuscript describes the role of periostin in hemodialysis vascular access stenosis at 1, 2, and 4 months post-implantation of polytetrafluoroethylene (PTFE) grafts in dogs.
Major Comments
1. It is unclear how the intima area is calculated from a longitudinal view. How far along the graft is considered arterial region given the graft is 8 cm long?
2. What is the rationale of choosing Fisher’s post hoc instead of Tukey’s tests? Fisher’s tests do not provide protection against false positives during multiple comparisons.
3. Most of the immunostaining images do not match the gene expression data. For example, Figure 2D says the gene expression of periostin is significantly increased in the middle region at 4 months when compared to 1 month and 2 months; however, the immunostaining images indicate the strongest positive signals at 1 month in Figure 2A.
4. It is unclear how explants were cut into three regions and processed for RNA isolation. Were they already paraffin-embedded tissue sections (line 262)?
5. There are six explants cut to three regions at each time point? Why are there more data points at 2 months and 4 months in Figure 6? This is critical because three of the seven figures are based on gene expression data. What could be the potential explanation for high correlations at all time points except periostin vs collagen at 4 months (Figure 6)? Why is periostin vs collagen correlated in the venous but not middle or arterial region (Figure 7)?
Minor Comments
1. What do I and A mean in Figure 2?
2. Figure 3B and Figure 3C are never mentioned in the Results.
3. What are the green arrows for in Figure 3?
4. Results section 2.4 seems like a discussion with four references. Please consider moving it from Results to Discussion.
5. Please add scale bars in Figure 4. Which month is Figure 4A-B representing?
6. What is the thickness of PTFE grafts? What are the inner diameters of the carotid artery and the jugular vein in dogs?
Author Response
May 01, 2020
International Journal of Molecular Sciences
Editorial Office
Dear Dr. Reviewer 2:
Thank you very much for your E-mail of April 23 regarding our manuscript “Possible roles of periostin in the formation of hemodialysis vascular access stenosis after polytetrafluoroethylene graft implantation in dogs” (Manuscript ID: ijms-787151).
Below please find our point-by-point responses to the reviewers’ comments, and the changes made in the manuscript are highlighted by the “Track Changes” function in Microsoft Word.
Our responses to you is as follows.
Comments and Suggestions for Authors 2
- As described in the methods section, the harvested PTFE grafts (length: 7-8 cm) were cut into three equal parts of between 2.3-2.6 cm each. The part near the arterial anastomotic side was called the arterial anastomotic region, and the part near the venous anastomotic side was called the venous anastomotic region. The remaining part that was remote from both anastomotic sides was called the middle region. We have added the definite lengths of each part in the text in the methods.
- As noted, use of Fisher’s post hoc test for multiple comparisons may result in increased false-positive results, but since the statistical tests were performed among 3 groups, we feel that the requirements for Fisher’s post hoc test were met.
- As noted, the periostin-positive staining area or color strength in Figure 2 was not very parallel to the gene expressions evaluated by RT-PCR. We feel that the quantitative evaluation of some matricellular proteins in immunostained images is very difficult. As is well known, the positive staining color strength changes with the thickness of the paraffin tissue section and the time for color development with AEC substrate. Therefore, it is difficult to accurately quantify the expression of these extracellular matrixes by color extraction analysis. In the present study, the aim was to present the characteristic distribution of periostin by immunostaining, and expressions were quantitatively analyzed by RT-PCR.
- Yes, we extracted RNA from Carnoy-fixed, paraffin-embedded tissue blocks. A total RNA isolation kit is now available on the market. We apologize for the lack of detail in the original methods. We have now provided a fuller description of the extraction procedure in the methods as follows (Lines 288-291):
To extract the RNA of the PTFE grafts, 10 sheets of the 20-μm-thick paraffin sections were collected from the respective Carnoy-fixed, paraffin-embedded tissue blocks using a microtome (LITORATOMU, REM-710, Yamto Koki Kogyo Ltd., Saitama, Japan). Total RNA was extracted in accordance with the protocol provided in the total RNA isolation kit (ISOGEN PB Kit, NIPPON GENE CO., LTD., Tokyo, Japan)
- As noted, the data points in Figure 6 at 1 month seem smaller when they are compared with the time points at 2 months and 4 months. We found that the expressions of periostin, collagen, and TGF-β1 in the PTFE grafts 1 month after PTFE graft implantation were not as evident, and the expressions of these factors in many samples were undetectable. Therefore, these data were overlapped at 0 blot sites.
- All expressions of periostin, collagen, and TGF-β1 in the PTFE grafts showed higher correlations at 1 and 2 months after implantation than at 4 months. Although the detailed reasons for this are not clear, we suspect that the hyperplastic intimal activity at the respective regions may be related to those correlations. For example, as shown in Figure 1 D, although the total intimal formation of PTFE grafts showed a time-dependent increase during the 4-month post-implantation period, a peak of intimal formation at the venous anastomotic region of the PTFE grafts was reached 2 months after implantation and maintained thereafter, indicating that the intimal proliferative factors were most active during the early phases (1 and 2 months), and more closer correlations may be seen between these factors than in the later phase (4 months).
- The detailed reasons why the correlation between periostin and collagen (Figure 7) was evident at the venous region, but not at the middle or arterial region are unclear. One of the possible reasons for this may arise from the weak correlation between periostin and collagen. As shown in Figure 5, although a significant positive correlation was observed between the expressions of periostin and collagen, the coefficient value (r) was 0.485, which is smaller than the others (r value between periostin and TGF-β1= 0.668; r value between TGF-β1 and collagen = 0.714). In these statistical analyses, the sample numbers for the calculations were much larger. However, the samples for the calculations in Figure 7 were divided into 3 parts, and, therefore, the statistically significant differences may have disappeared due to the decreased sample numbers in these groups.
Minor concerns:
1) As suggested, we have defined I and A in Figure 2 as follows: I, intimal side; A, adventitial side.
2) As suggested, we have added following sentences to illustrate Figure 3B and Figure 3C.
Line (114-115): A similar immunostaining pattern for TGF-β1 was also found 2 and 4 months after PTFE graft implantation (Figure 3B and 3C).
3) As suggested, we have added the meaning of the green arrows in the legend of Figure 3 as follows:
Green arrows: TGF-β1-positive cells.
4) As suggested, we have modified the Results section 2.4 to the following (Lines 127-139):
Figure 4A shows the immunohistologic examinations of the serial sections from the venous anastomotic regions with antibodies against vimentin and α-SMA. These loci were matched to the yellow frames of Azan-Mallory staining, as shown in Figure 4B. The intima areas in the venous anastomotic region were occupied mainly by α-SMA-positive cells, and the immunostaining for vimentin in the serial section showed that most of these α-SMA-positive cells were overlapped with the vimentin-positive cells. On the other hand, as can be seen in the high magnification views of vimentin and α-SMA immunostaining, although vimentin-positive staining was found in all cellular components at these venous regions (including either in the intima area and in the splits of PTFF material wall), some cellular components in the border zone between the intima and the PTFF material wall, as well as in the splits of PTFF material wall, were negative for α-SMA (yellow arrows, Figure 4A). A blue color on Azan-Mallory staining indicates collagen distribution in the venous anastomotic region, suggesting that collagen also constitutes part of the intima component. Figure 4C shows the expression of collagen I mRNA in three different regions 1, 2, and 4 months after PTFE graft implantation. Collagen I expressions were time-dependently increased in all three regions.
In addition, we have moved the discussion-like sentences to the Discussion section, as follows (Lines 185-199):
On the other hand, the intima areas in the venous anastomotic region were occupied mainly by α-SMA-positive cells, and the immunostaining for vimentin in the serial section showed that most of these α-SMA-positive cells were overlapped with the vimentin-positive cells, indicating that these cells originated from mesenchymal progenitor cells [15]. As is well known, mesenchymal cells contain mainly fibroblasts with the phenotypic type of myofibroblasts, and since the majority of intima component cells were α-SMA-positive, it was concluded that the largest cellular components in the proliferative intima after implantation of PTFE grafts in dogs were myofibroblasts [16]. On the other hand, as can be seen in the high magnification views of vimentin and α-SMA immunostaining, although vimentin-positive staining was found in all cellular components at these venous regions (including either in the intima area and in the splits of PTFF material wall), some cellular components in the border zone between the intima and the PTFF material wall, as well as in the splits of PTFF material wall, were negative for α-SMA (yellow arrows, Figure 4A), indicating that these cells were fibroblasts. Therefore, myofibroblasts were a major cellular component, and fibroblasts comprised only a small proportion in the hyperplastic intima. Although fibroblasts seemed not likely to be a major cellular component within the hyperplastic intima, our previous studies showed that these cells were important to migrate from one place to others [17, 18].
5) As suggested, we have added scale bars to Figure 4.
6) The thickness of the PTFE graft wall is 0.6 mm, and the outside diameters of the canine carotid artery and jugular vein under anesthesia were 4-6 mm and 6-8 mm, respectively.
We appreciate the helpful comments we received and believe that all these corrections have made our paper much more valuable.
Thank you very much for your kind consideration.
We look forward to the appearance of our revised manuscript in International Journal of Molecular Sciences.
Sincerely yours,
Denan Jin, M.D., Ph.D.
Department of Innovative Medicine, Graduate School of Medicine, Osaka Medical College, 2-7 Daigaku-machi, Takatsuki, Osaka 569-8686, Japan.
TEL: +81-72-683-1221 (Ext2141)
FAX: +81-72-684-6730
E-mail: pha012@osaka-med.ac.jp

Round 2
Reviewer 1 Report
The authors have addressed all the comments.
Minor comments:
Please proof-read the manuscript for syntax errors.
Author Response
May 02, 2020
International Journal of Molecular Sciences
Editorial Office
Dear Dr. Reviewer 1:
Thank you very much for your E-mail of May 02 regarding our manuscript “Possible roles of periostin in the formation of hemodialysis vascular access stenosis after polytetrafluoroethylene graft implantation in dogs” (Manuscript ID: ijms-787151).
In accordance with your instruction, we have red our manuscript carefully. On the other hand, we corrected our manuscript and the revised contents of this paper by the native speaker of English before the submissions.
Thank you very much for your kind consideration.
We look forward to the appearance of our revised manuscript in International Journal of Molecular Sciences.
Sincerely yours,
Denan Jin, M.D., Ph.D.
Department of Innovative Medicine, Graduate School of Medicine, Osaka Medical College, 2-7 Daigaku-machi, Takatsuki, Osaka 569-8686, Japan.
TEL: +81-72-683-1221 (Ext2141)
FAX: +81-72-684-6730
E-mail: pha012@osaka-med.ac.jp
Reviewer 2 Report
Please correct the following typographical errors: "PTFE" (line 137),"PTFE" (line 137), "PTFE" (line 138), "PTFE" (line 197), "PTFE" (line 198), "PTFE" (line 198), from one place to "another" (line 202). Thank you.
Author Response
May 02, 2020
International Journal of Molecular Sciences
Editorial Office
Dear Dr. Reviewer 2:
Thank you very much for your E-mail of May 02 regarding our manuscript “Possible roles of periostin in the formation of hemodialysis vascular access stenosis after polytetrafluoroethylene graft implantation in dogs” (Manuscript ID: ijms-787151).
We are so sorry our careless mistakes.
We have corrected typographical errors indicated by Reviewer 2
Lines; 134,145, 194,195 of PEFF to PTFE.
Thank you very much for your kind consideration.
We look forward to the appearance of our revised manuscript in International Journal of Molecular Sciences.
Sincerely yours,
Denan Jin, M.D., Ph.D.
Department of Innovative Medicine, Graduate School of Medicine, Osaka Medical College, 2-7 Daigaku-machi, Takatsuki, Osaka 569-8686, Japan.
TEL: +81-72-683-1221 (Ext2141)
FAX: +81-72-684-6730
E-mail: pha012@osaka-med.ac.jp